# Co-Operation between Aneuploidy and Metabolic Changes in Driving Tumorigenesis

**DOI:** 10.3390/ijms20184611

**Published:** 2019-09-18

**Authors:** David L. Newman, Stephen L. Gregory

**Affiliations:** 1Department of Molecular and Biomedical Science, University of Adelaide, Adelaide 5006, Australia; david.newman@adelaide.edu.au; 2Genetics and Molecular Medicine, College of Medicine and Public Health, Flinders University, Adelaide 5042, Australia

**Keywords:** aneuploidy, metabolism, oxidative stress, reactive oxygen species, glycolysis, DNA damage

## Abstract

Alterations from the normal set of chromosomes are extremely common as cells progress toward tumourigenesis. Similarly, we expect to see disruption of normal cellular metabolism, particularly in the use of glucose. In this review, we discuss the connections between these two processes: how chromosomal aberrations lead to metabolic disruption, and vice versa. Both processes typically result in the production of elevated levels of reactive oxygen species, so we particularly focus on their role in mediating oncogenic changes.

## 1. Introduction

Aneuploidy, an abnormal set of chromosomes, was first identified as a biomarker for epithelial cancer in the late 1800s [1], and remains a typical feature of advanced cancers, with more than 80% of solid tumours showing visible chromosomal abnormalities [2]. It has also been known for many decades that tumours typically show altered patterns of energy use, particularly an elevated rate of glycolysis [3]. Recent work has built up a picture of tumour metabolism that is complex and varied, but still strikingly abnormal [4]. With such a long history of work on these two stereotypical features of cancer, it is perhaps surprising that there has not been intense scrutiny of the links between them. In part that is probably because aneuploidy has been seen as a stochastic, highly variable process, with unpredictable outcomes depending on which chromosomes are affected. More recently, however, it has become apparent that there are some reproducible effects of aneuploidy regardless of the exact aberration [5]. These effects of aneuploidy include oxidative stress, prompting a new interest in the metabolic impact of aneuploidy. In this review, we examine the connections between aneuploidy and metabolism in driving tumorigenesis, with a particular focus on the role of oxidative stress.

## 2. Metabolic Changes Leading to Tumorigenesis

Metabolic pathways interact with almost every oncogenic process, but there are three areas in which the contribution of metabolic mutations to oncogenesis has been particularly well characterized: the elevation of glycolysis, the generation of oncometabolites and the production of reactive oxygen species [6,7].

Glycolysis was the first metabolic process known to be aberrantly elevated in cancer. This Warburg effect was initially interpreted as a shutdown of oxidative phosphorylation that was oncogenic in itself [8]. However, we now know that oxidative phosphorylation generally continues unabated even in highly glycolytic cancer cells, so alternative explanations for the high rate of glycolysis have been proposed. The argument that glycolysis is a more rapid source of energy is not compelling – normal tissues that must respond to rapid energy demands, such as muscles, do not prioritize glycolysis. However, it may be that membrane-localized demands for ATP can be most efficiently met by glycolysis [9]. The opposite suggestion has also been made: that rather than needing more ATP, some cancers need to deplete their cytoplasmic ATP in order to avoid inhibiting glycolysis [10]. In this model, glycolysis is critical for providing biosynthetic precursors for nucleotide and amino acid synthesis that are rate limiting for proliferation. Elevation of glycolysis in transformed cells has also been proposed as an important part of surrounding normal cells’ efforts to eliminate them from epithelia [11].

A problem with some of these models is that we do not know what the flux is through each of the possible output pathways from glycolysis, and they can be mutually exclusive. For example, every molecule of lactate produced is also one molecule of glucose that was not made into a nucleotide. However, despite the need for nucleotides, production of lactate is frequently elevated in cancers. Recent flux analysis of glycolysis in cancer cells suggests that a surprising fraction of glycolysis is ‘wasted’ as lactate, which is secreted, then used as fuel by other cells [12,13,14]. However, although it is something of a dead end metabolically (Figure 1), lactate production has potential benefits in promoting autophagy [15], as well as modulating the immune environment of the tumour [16]. However, the production of lactate restricts the use of glucose to refill the Tri-Carboxylic Acid (TCA) cycle, an essential step in generating citrate for lipid synthesis. Flux analysis has shown that this can explain the dependency on glutamine, a key fuel in many cancers that generates TCA cycle intermediates, lipids and nucleotides [17,18,19].

Glucose can also be used to produce reducing power in the form of NADPH via the pentose phosphate pathway. NADPH facilitates lipid synthesis, and may be critical for maintaining redox homeostasis, as will be seen when we consider the production of reactive oxygen species. It is worth noting that the demand for new nucleotides (via the pentose phosphate pathway) will lead to a build-up of NADPH that will block the first step in the Pentose Phosphate Pathway (PPP) unless the NADPH is used up. This step is tightly regulated, being induced by Ras and PI3-kinase and inhibited by p53 and AMPK among many others that are typically altered in tumorigenesis to promote this pathway [20]. Since proliferating cells need both new nucleotides and new lipids, the expectation is that glucose use is balanced between the PPP pathway and the production of citrate for fatty acid synthesis that consumes NADPH [21]. These demands for lipids and nucleotides, as well as the need for additional amino acids for protein synthesis, are likely to be major driving forces behind the Warburg effect (elevated use of glucose without elevated mitochondrial output). While this is not the most efficient way to generate ATP, it allows the mitochondria to use alternative fuels, such as glutamate, with the possible benefit of generating oncometabolites and elevated reactive oxygen species (ROS) levels [21]. Tumours that produce elevated levels of reactive oxygen species are potentially able to drive nucleotide synthesis harder as NADPH can be rapidly removed in reducing spent antioxidants, such as glutathione and catalase. Some tumours, such as KRAS-driven adenocarcinomas, can work around the dependence on NADP by promoting the generation of nucleotides via the non-oxidative arm of the PPP pathway [22]. However, even in this case, flux through the oxidative arm is still high, reflecting the demand for nucleotides that has led to therapy efforts targeting nucleotide synthesis [23]. Assessing the metabolic reprogramming of multiple cancer types has suggested that the intersection of glycolysis with one-carbon metabolism may be a fruitful therapeutic angle, presumably again because of their contribution to nucleotide synthesis [14].

The metabolic changes described to this point could be considered enabling changes that potentiate high rates of proliferation, rather than specifically driving oncogenesis. However, there are other metabolic changes that can specifically promote cellular transformation [4]. Mutations that lead to the overproduction of 2-hydroxyglutarate, fumarate and succinate are associated with tumour progression in a range of cancers [24,25]. The mechanism of action of these oncometabolites is not simple, but includes inhibition of enzymes such as DNA or histone demethylases that causes genome-wide epigenetic changes to gene expression [26], as well as aberrant stabilization of HIF-1alpha that promotes glycolysis and vascularisation [27] and a range of effects on inflammatory signalling [25]. Mutations in metabolic enzymes such as IDH, FDH and SDH that generate these oncometabolites are not as pervasive in cancer as the elevation of glycolysis, but are clear driver mutations in some tumour types, particularly gliomas and some leukemias [24]. Paracrine-acting oncometabolites, such as lysophosphatidic acid, drive proliferation and pro-tumour differentiation of nearby stromal fibroblasts [28]. Several metabolic disruptions including hypoxia, oxidative stress [29] and energy depletion are implicated in oncogenesis as they normally trigger the p53-dependent nucleolar surveillance pathway to arrest the cell cycle. Cancers have long been known to subvert this response [30]; therefore, clinical trials are under way for drugs that independently trigger the nucleolar response [31].

The elevated production of reactive oxygen species (ROS) is a common metabolic change seen in almost all cancer types, and is currently a focus for metabolic therapies [4,32,33]. Reactive oxygen species is a broad term including rapidly reacting damaging molecules such as hydroxyl radicals, as well as less reactive species, such as hydrogen peroxide, that primarily react harmlessly with antioxidant enzymes [34]. The ROS can arise from a wide range of sources, including hypoxic respiration inside poorly vascularised tumours, mutations in metabolic genes such as IDH, activation of NADPH oxidases and, as will be discussed later, aneuploidy. Although ROS have often been considered deleterious to cells due to their damaging effect on DNA, proteins and lipids, it has become clear that they can promote oncogenesis. The most obvious mechanism is by raising the rate of mutation: in order to change from a normal epithelial cell to an invasive, metastatic, apoptosis-resistant cell, cancers must accumulate numerous mutations, so an elevated rate of DNA damage can speed-up that process. There are many other potential benefits for cancer cells with elevated ROS: there are ROS-responsive enzymes that control DNA damage repair (ATM), hypoxia (PHD) and the rate of glycolysis (PDHK), among others [35,36,37]. As previously mentioned, ROS also tend to deplete antioxidant stores that are replenished using NADPH, which allows more use of the pentose phosphate pathway to make nucleotides. Tumours may rely on a certain level of ROS to maintain their high-proliferation metabolic state [38], but this is always a delicate balance, as too much oxidative stress will kill even hardy apoptosis-resistant cancer cells.

## 3. The Effect of Aneuploidy on Tumorigenesis

Disruption to the genome is a hallmark of cancer, and in the vast majority of cases, this is not just point mutations or minor indels, but extends to the point at which changes to the karyotype can be seen [2,39]. These chromosomal changes could just reflect the damage accumulated by an aging and aberrant cell. However, it has become clear that aneuploidy is not just a symptom of cancer; it also has a causative role. One of the most striking experiments to show this simply blocked cytokinesis for one cell division, then showed that these tetraploid cells produced cancers in a mouse xenograft, while the diploid controls did not [40]. More recently, fly models have shown that induction of aneuploidy in an epithelium drives dysplasia and over proliferation of stem cells [41], and even loss of a single chromosome can cause transformation of tetraploid MEFs [42]. In this context, it is interesting to note that many studies have found that the initial response to aneuploidy is often a p53-dependent decrease in proliferation, particularly if the aneuploidy is drastic [43,44]. This is usually temporary, however, with subsequent increases in proliferation thought to come from mutant subclones that escape the normal cellular responses to genetic aberration. Trisomy 21 is an interesting exception to this, as it incurs much lower risk of most solid tumours and increased risk, mainly, of childhood leukemia [45]. This may be due to the relatively small, specific set of genes misregulated when one chromosome is stably altered [46].

The role of tetraploidy in promoting oncogenesis has been clinically validated in Barrett’s oesophagus, in which a genome doubling is a well characterized step towards adenocarcinoma in precancerous lesions that have lost p53 [47]. Similar results have recently been found for lung cancer evolution [48]. More generally, genetic diversity is a poor prognostic indicator for almost every cancer type [39,49,50]. While an aberrant karyotype may reflect the age and mutational burden of the progenitor cells, some specific chromosomal rearrangements can be oncogenic in their own right. The most obvious example is the high frequency of specific translocations in leukemias, such as the Philadelphia chromosome in chronic myeloid leukemia. In this case, a pro-proliferative gene (ABL) loses its autoinhibitory domain in the translocation, and the result is a potent oncogene [51]. Loss of tumour suppressors by deletion events are even more common, with deletion of 17p and the consequent loss of p53 being a poor prognostic marker in many cancer types [52].

It should be noted that a change from stable diploidy to a stable aneuploidy is not the normal course of cancer evolution. Instead, it is common to find ongoing chromosomal instability (CIN) that generates significant subclonal diversity of the chromosome number in tumours [49]. Given that chromosomal instability can be induced by such relatively minor changes as disruption of the normal epithelial architecture [53], it is no surprise that CIN is an early and pervasive feature of tumorigenesis. There are several benefits of CIN for tumours: it varies them over time [54], contributing significantly to their chance of becoming metastatic [55] and resistant to chemotherapy [56,57]. Extensive transcriptional [5] and proteomic analyses [58,59] in human and yeast models have shown that CIN cells are significantly different to stable diploids. These effects of chromosomal instability are relatively uncontroversial; less clear is whether CIN makes tumours vulnerable to immune attack. Initial reports suggested that the changes brought about by CIN were likely to produce neo-epitopes that could make tumours more susceptible to immunotherapy [60]. This was supported by evidence that DNA damage and mitotic errors led to an inflammatory response via the cGAS-STING pathway [61,62] and the production of inflammatory cytokines [63]. Ectopic activation of this pathway can cause tumour regression [64]. However, activation of this auto-immune response does not necessarily assist with either normal tumour clearance or with immunotherapy, as the cGAS-STING pathway may promote metastasis [55] and highly aneuploid tumours have been found to attract less cytotoxic T cells and show poorer responses to immunotherapy, at least in melanoma [65]. This may be because high levels of aneuploidy reflect a highly mutated, late-stage cancer that must have evolved immune resistance to reach that stage. So the fact that aneuploidy tends to trigger an inflammatory response may mean that cancers that can tolerate aneuploidy are likely to be resistant to, or even dependent on, the inflammatory response [66,67]. Part of the resistance may be via the secretion of lactate into the tumour microenvironment, which is detrimental to a range of immune cells [7,68]. Dependence on inflammation commonly occurs in cancers that evolve in a pro-inflammatory environment and can use normally protective signals such as TNF and IL-10 to promote their proliferation and metastasis [69,70]. Therefore, it appears that aneuploidy is likely to stimulate an inflammatory response, but whether the response is beneficial or deleterious can vary significantly depending on the tumour.

Having briefly reviewed the effects of metabolism and aneuploidy on tumorigenesis, we now move to consideration of how these two processes are related.

## 4. Effects of Aneuploidy on Metabolism

Aneuploid cell populations can exhibit heterogeneous phenotypes due to the wide range of possible gene dosage alterations [71]. The ensuing cellular stress responses, however, appear to be more homogeneous. These include (but are not limited to) upregulation of DNA repair mechanisms, autophagy/lysosomes, and antioxidant levels as observed in yeast and fly models [58,72,73]. These are required in order to compensate for aneuploidy-associated metabolic stresses, such as proteotoxic burden, overloaded endoplasmic reticulum (ER), increased glycolytic flux and mitochondrial activity, and an increase of reactive oxygen species (ROS) [73,74,75,76]. These stresses contribute to the metabolic and proliferative defects initially found in aneuploid yeast [77].

Aneuploidy alters several metabolic pathways compared to diploid controls [78,79]. Both stable aneuploid and CIN cells alter mitochondrial numbers and activity, typically resulting in an increase of ROS in mouse and fly models [76,80,81]. Build-up of ROS leads to pathological consequences such as double-strand breaks (DSB) in DNA, protein oxidation, lipid damage, mitotic catastrophes, and has been reported to alter expression of tumour suppressor genes and oncogenes [82,83,84]. Excessive oxidative stress affects key regulatory proteins involved in mitotic progression, which likely contributes to the slower proliferation seen in aneuploid cells. Cdc25 phosphatases, for example, regulate commitment to mitosis and contain active cysteine residues, which create inhibitory disulphide bonds in the presence of H_2_O_2_, thus preventing mitosis [85]. One study showed antioxidant treatment partially restoring proliferation rates in MEFs with induced CIN, which highlights the negative effects of ROS on mitosis [82]. Therefore, in order for genetically unstable cells to tolerate high levels of ROS and to overcome mitotic safeguards against redox imbalances, we would expect a corresponding increase in antioxidant gene expression [86,87]. Indeed, a proteomic study using disomic yeast revealed upregulation of genes involved in protection against oxidative stress [58].

The increase in glycolysis and mitochondrial activity may be partly to accommodate the energy deficit caused by the additional protein synthesis, folding and clearance required in a high-ROS environment [75,78,88]. The increased mitochondrial activity could also stem from poly (ADP-ribose) polymerase (PARP) activity, which is upregulated in CIN cells in the fly model [89]. PARP is activated by DNA damage and its activity depletes NAD+, resulting in mitochondrial stress [90]. Given the elevated mitochondrial activity and ROS levels, genetically unstable cells have an increase in damaged mitochondria that require removal [73,78]. Unsurprisingly, upregulation of mitophagy in such cells has been shown to result in decreased ROS and cell death in fly and vertebrate models [73,91].

Some of the above-mentioned stresses likely stem from gene dosage changes imposing a proteomic burden on aneuploid cells [58,75], although work in yeast has shown that, to some extent, cells can buffer protein dosage [92]. Additional protein synthesis, folding and degradation place extra demands on the ER and cellular chaperones, such as heat shock protein 90 (HSP90). Consequently, aneuploid cells are sensitive to inhibition of HSP90′s protein-folding ability [93]. Likewise, peptides that do not have protein partners at the correct stoichiometry require clearance or risk aggregating [94]. Indeed, CIN is marked by an increase in lysosomal activity, and may push lysosomal degradation pathways beyond their limits [78]. To some extent, protein aggregation serves as a dosage compensation mechanism for the overburdened chaperone, proteasome, and lysosomal apparatus [94]. Although degradation was found to be the major contributor to the removal of excess proteins, aggregation was also a significant factor. This may explain the results of another recent yeast study that unexpectedly saw overexpression of balanced sets of proteins disrupting proliferation of aneuploid cells more than those with additional unbalanced proteins [95]. This implied that extra copies of full sets of interacting protein complexes exert more stress on cellular functions than additional unpartnered proteins that will aggregate and be sequestered away.

Sustained stress on the ER from misfolded proteins can also result in triggering unfolded protein response (UPR) mechanisms [88,96]. Prolonged activation of the UPR comes with its own set of consequences, such as calcium release from the ER, which can trigger overactivity of the Krebs cycle, loss of mitochondrial integrity, increased ROS, release of cytochrome C, and cell death [97]. In yeast, aneuploidy may solve its own problem of ER stress by increasing the dosage of the appropriate ER processing enzymes [98]. However, as well as the issue of gene dosage, aneuploidy-induced oxidative stress can also contribute to the formation of protein aggregates. This was demonstrated in a recent study in flies that showed significant decreases in protein aggregation in chromosomally instable (CIN) cells when a common antioxidant, catalase, was overexpressed [89]. Therefore, increased ROS levels could contribute to the formation of protein aggregates, putting more stress on the ER, which results in more ROS, creating a feedback loop that should result in the death of defective cells (Figure 2). A similar destructive feedback loop has been proposed for DNA damage, which promotes aneuploidy, which elevates ROS, which increases DNA damage [99,100]. It appears that aneuploid cells use ROS to enhance relatively minor alterations in ploidy into a significant damage signal that disrupts protein folding and DNA homeostasis, as well as mitochondrial metabolism

## 5. Evidence for How Metabolism can Impact Aneuploidy Generation or Tolerance

Aneuploidy is readily generated by any flaw that results in chromosome segregation before complete decatenation of the replicated DNA or before proper amphitelic attachment to the mitotic spindle has been established [101]. However, as cancer is now increasingly recognised as a metabolic disease [102], recent evidence has suggested that disturbances in energy metabolism can play a part in the generation of aneuploidy, particularly via the generation of ROS. A pathological build-up of ROS—stemming from an initial disruption of cellular metabolism—can come about in many ways. Some examples include mutations in oncogenic KRAS, resulting in the upregulation of mitochondrial metabolism followed by a build-up of ROS [103]. Similarly, upregulation of manganese superoxide dismutase (MnSOD) results in mitochondrial damage via the increase of H_2_O_2_ production, which causes sustained activation of AMP-activated kinase (AMPK) [104]. Nutrient deprivation has also been shown to induce ROS through AMPK [105]. In some tissues, low mitochondrial ATP output can result in ROS from mitochondrial NADPH oxidase 4 [106]. Additionally, NADPH oxidases 1 and 4 also produce H_2_O_2_ in response to growth factors [85]. Hypoxia has also been shown to be a driver of ROS generation within poorly vascularised solid tumours [107,108].

The accumulation of ROS is thought to impact aneuploidy primarily by causing double-strand breaks in DNA, which increases the frequency of mitotic errors [109]; however, ROS can also impact the generation of aneuploidy in other ways (Figure 3). For example, H_2_O_2_-treated yeast have mis-localisation of Bub1 from kinetochores, resulting in the inactivation of the SAC and premature exit from mitosis [110]. Exogenous H_2_O_2_ also causes centrosome amplification [111], which often results in aneuploidy [112]. Similarly, ROS associated with type 2 diabetes increased the likelihood of centrosome amplification via disruption of centrosomal ROCK1 [113]. ROS has also been reported to disrupt the mitotic machinery, for example, by oxidation of Aurora A and B, increasing the occurrence of misaligned chromosomes and chromatin bridges, resulting in aneuploidy [114,115]. Even low levels of chronic ROS can result in oxidation of DNA within the GGG triplet found in the telomeres, resulting in telomere shortening and an increase in chromosome instability [116]. However, oxidation of mitotic components is not the only means of metabolic disruptions inducing genomic instability. Studies investigating energy metabolism highlight how defective coenzyme A functionality (an essential cofactor for the metabolism of carboxylic acids and lipids) in *Drosophila* and fission yeast leads to genomic instability [117,118]. This is likely due to hypoacetylation of histones, which results in defective repair of damaged DNA [118]. Research surrounding sirtuins (SIRT), which are known for their NAD+-dependent deacylase activity, has shown them to be critical for cellular metabolism, genomic stability, and preventing build-up of pathological ROS [119]. Deletion of the mitochondrial SIRT3 results in an increased level of superoxide and genomic instability and enhances tumour growth in mammary glands [120]. Similarly, loss of SIRT4, another mitochondrial sirtuin, allows for continued glutamine entry into mitochondria, thereby preventing cellular arrest following DNA damage, resulting in the accumulation of genomic damage [121].

While disruptions to metabolism that generate oxidative stress promote aneuploidy by increasing the error rate in mitosis, metabolic changes can also select for aneuploidy. Cells that are metabolically damaged may be benefitted if a mitotic error leads to a favourable chromosomal alteration. For example, yeast carrying a thiol peroxidase deficiency gained spontaneous duplications in chromosome 11 [122]. Thiol peroxidases are highly conserved antioxidants that are the first line of defence against ROS-induced DNA damage. The specific chromosomal duplication resulted in significantly elevated expression of CCP1 and UTH1, which encode mitochondrial genes that protect against oxidative stress and rescue proliferation in cells lacking peroxidase [122]. We have discussed above how aneuploidy frequently results in an increase of ROS, but it should be noted that this study shows how specific chromosomal amplifications can also protect against oxidative stresses. A similar yeast study has shown duplication of chromosome 2 counteracting proteotoxic stress, another hallmark of aneuploid cells [98]. In this case, the specific aneuploidy was associated with an increased expression of ALG7 and PRE7, which are important for protein maturation at the endoplasmic reticulum (ER) and proteasome function, respectively. Overexpression of ALG7 and PRE7 protected against tunicamycin-induced ER stress in UPR-deficient cells, suggesting that these genes induced ER stress resistance independently of the UPR [123]. Loss of chromosomes can also be adaptive: a series of studies report on diploid yeast becoming monosomic for chromosome 8 in order to overcome a telomerase deficiency [124,125]. These examples demonstrate how specific increases in gene dosage can confer adaptive countermeasures for overcoming stresses brought about by genetic disruption.

## 6. Conclusions

It is no longer enough to know that tumours are likely to be aneuploid and have a disrupted metabolism. Specific genetic and metabolic changes have become commonplace markers for cancer prognosis, so these processes clearly contribute to driving tumour progression and significantly impact the likelihood of effective chemotherapy. We are beginning to understand how the genome and metabolome are interlinked, particularly via the generation and clearance of ROS. A recurring theme is that metabolism can act as an amplifying sensor for genetic alterations, reacting to relatively minor changes in gene dosage by generating feedback loops that should promote further cellular disruption, leading to cell death. While cancers may hijack these protective responses and avoid cell death, they nonetheless represent a point of significant difference between the tumour and normal somatic tissue. There are ongoing efforts to target both genetic and metabolic differences in cancer cells as a chemotherapeutic approach [6,126,127]. We cannot expect to prevent DNA damage or aneuploidy, and most tumours will be adapted to their new metabolism by the time we identify them, so our job is to find where the constraints on a cancer cell bite: is it their ability to tolerate more ROS, or their sensitivity to any further demands on mitophagy, or their sensitivity to protein overexpression or further ER stress? To facilitate this, more understanding is needed, particularly of how aneuploidy is sensed in the nucleus, methods to identify tumours that are close to their oxidative stress limit and finding ways to target metabolism that will preferentially affect genetically disrupted cells.

## Figures and Tables

**Figure 1 ijms-20-04611-f001:**
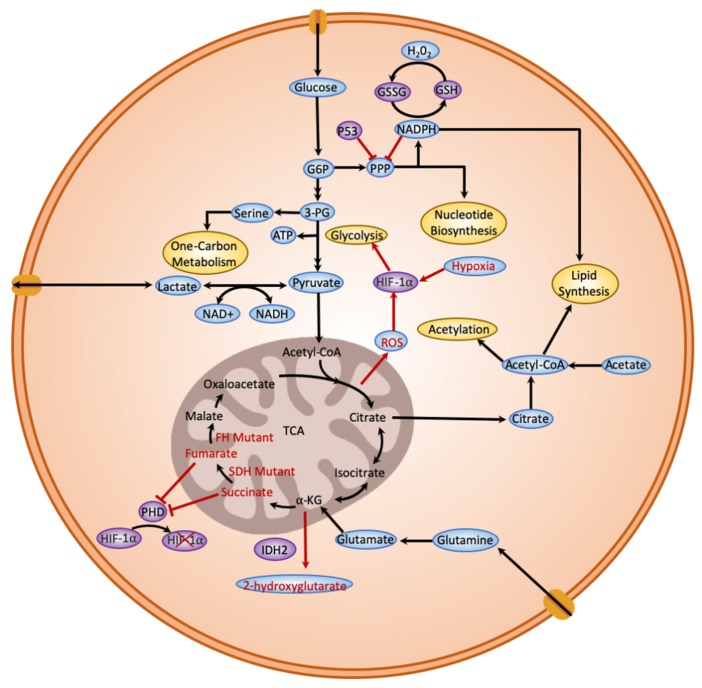
A schematic depicting the Warburg effect in cancer cells. The diagram illustrates the distinct aspects of the Warburg effect in cancer cells, containing glycolysis, pentose pyruvate pathway, lactate fermentation, glutamine metabolism, reactive oxygen species (ROS) generation, Tri-Carboxylic Acid (TCA) cycle, intermediates from the TCA cycle to synthesize lipids, and use of mutations in the TCA cycle (highlighted red) to synthesize oncometabolites. Important metabolic pathways are highlighted in yellow and important enzyme-regulating steps in glycolysis are highlighted in purple. Red lines with blunt ends indicate an inhibitory mode of action.

**Figure 2 ijms-20-04611-f002:**
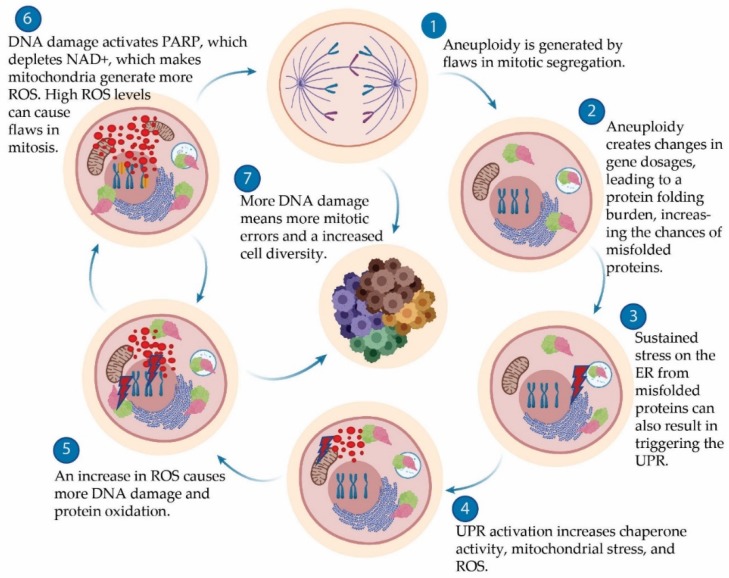
Overview of chromosomal instability (CIN) effects on metabolism. (**1**) The onset of aneuploidy brings about several cellular consequences including (**2**) proteomic imbalances [50], which can lead to (**3**) endoplasmic reticulum (ER) stress and activation of unfolded protein response (UPR) mechanisms [89]. This leads to (**4**) increased mitochondrial activity due to increased energy demands and ROS production, which brings about (**5**) more DNA damage and protein oxidation. These lead to stress and damage to mitochondria, the ER and DNA. (**6**) DNA damage is repaired by poly (ADP-ribose) polymerase (PARP) activity, which uses NAD+, putting more stress on mitochondria, which results in more ROS [81]. (**7**) More DNA damage results in more mitotic errors, which increases cellular diversity.

**Figure 3 ijms-20-04611-f003:**
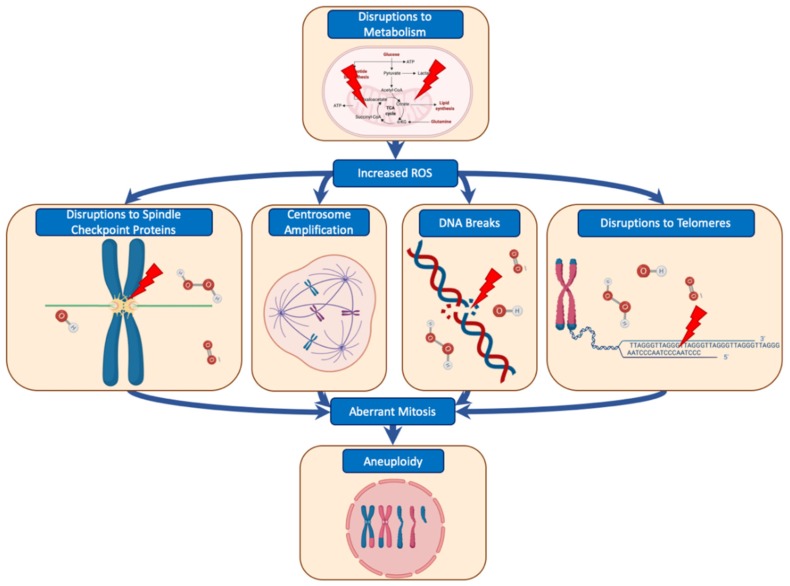
Metabolic disruption can cause aneuploidy. Disturbances in energy metabolism lead to a buildup of pathological ROS. This can play a part in aberrant mitosis via several ways, such as disruptions to spindle checkpoint proteins [101,102], centrosome amplification [104,105], DNA breaks [101], and disturbances to telomeres [108].

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
