# Peer review of "Co-Operation between Aneuploidy and Metabolic Changes in Driving Tumorigenesis"

_ijms, 2019, doi:10.3390/ijms20184611_

Round 1

Reviewer 1 Report

paper  well structured and sounding

text must be reviewed for some typos

additional consolidated data regarding modifications of some metabolic pathways and nuclear / nucleolar functions (lysophosphatidic acid, NORs, etc.)  should be added to complete the paper (i.e., see papers from Mazzocca A, or Cai S, or Bufo P, or Lewinska A, et more)

the final section "Summary" sounds better "Conclusions"

the funding institution is not cited, in despite of the disclaimer about the absence of any influence of it on the text content

Author Response

We have removed any typos that we could find.

We have discussed the relevance of LPA and the nucleolus as requested (lines 98-103), including papers by Mazzocca, Bufo, and Lewinska. 

The final heading has been changed to Conclusions as requested (line 322).

We have indicated our institutional affiliations at the top of the paper; no other organizations funded this work, which we have made clearer (line 342)

Reviewer 2 Report

The review by Newman and Gregory charts the links between chromosome dosage imbalances and metabolic disturbances in cancer. I believe that the review is timely – as they note, despite the abundance of research in this field, there has not been a significant attempt to synthesize it yet. In general, the review is well-written and informative, and I support its publication.  I’d ask the authors to address the following points:

1) Our current understanding of the Warburg effect is that it serves to generate cellular building blocks at the expense of lower ATP production (e.g., Vander Heiden, Science 2009). The authors mention this as a possible explanation: “glycolysis is critical for providing biosynthetic…”. However, I think that this explanation should be expanded.

2) In several places, the authors gloss over differences between aneuploid and CIN cells. For instance, they write: “One study showed antioxidant treatment partially restoring proliferation rates in aneuploid MEFs.” The cited study was actually done in SAC-deficient CDC20-mutant MEFs. It’s difficult from this experiment to know whether it was the instability or the aneuploidy that was buffered by antioxidant treatment.  A second example: “Aneuploid cells increase mitochondrial numbers and activity, which results in an increase of ROS.” The authors then cite one of their own papers using mad2-knockdown flies. The authors should clarify in the text whether they’re referring to experiments done with CIN cells or with “stable” aneuploidies.

3) Similarly, the authors sometimes gloss over differences between experiments conducted in yeast, flies, and mammalian cells. Readers may wish to know what system was used for a particular finding. For instance, the authors write, “This was demonstrated in a recent study that showed significant decreases in protein aggregation in chromosomally instable (CIN) cells when a common antioxidant, catalase, was overexpressed.” They may wish to change this to “in a recent study in flies” or something similar.

4) The BCR-ABL fusion is oncogenic due to the expression of the ABL kinase, and not for any reason related to aneuploidy or gene dosage imbalances. I believe that discussing this translocation in the context of aneuploidy-induced tumorigenesis is confusing at best. I would suggest that the authors eliminate these sentences, or otherwise clarify how this translocation is different than the other examples presented here.

5) The relationship between CIN, aneuploidy, and oncogenesis is more complex than described by the authors. In particular, in some instances, aneuploidy and CIN seem to have tumor suppressor properties: Weaver, Cancer Cell 2007, Baek Nature 2009, Silk PNAS 2013, Sheltzer Cancer Cell 2017, etc. This finding is worth discussing as well.  Could it be related to metabolism? One could imagine that CIN or aneuploidy-induced ROS elevation triggers DNA damage and causes a p53-dependent senescence phenotype, which blocks tumor growth. Or something else?  

6) Minor typo: “disturbances in energy metabolism can to play a part in the generation of aneuploidy.”

Once the points are addressed, I believe that the review will be fit for publication.

Author Response

1. We have expanded the description of the Warburg effect as a generator of biosynthetic precursors as requested (lines 74-78).

2. We have clarified whether we are talking about aneuploidy or instability in the manuscript (lines 189-190, 198-199, 206, 209-210)  In most cases it is hard to be definitive about what causes the phenotype (aneuploidy or aneuploidy plus instability), but we have now at least pointed out whether they are unstable or not.

3. We have put in more indications of the models used. It is perhaps our bias to think that the different models strongly agree about the effects of aneuploidy, but we accept that it can be interesting to know which system was used. This has been clarified (lines 137, 165, 192, 196, 210, 215, 219, 222, 231, 239, 243, 288)

4. We agree that the BCR-Abl fusion example does not show the effect of gene dosage, but it is a common type of aneuploidy that drives oncogenesis. We have made it clearer (line 145) that the translocations and deletions in that section are specific genetic aberrations that are distinct from the stereotypical effects of whole chromosome gain or loss.

5. We agree that the effect of aneuploidy on oncogenesis is a complex and controversial issue. We did not focus on the studies that showed reduced proliferation in aneuploid cells simply because we know that cancers get around this issue (e.g. they lose p53 etc). Nonetheless, we agree that this finding should be mentioned and have added it (lines 139-146).

6. The typo has been fixed.